Wedelolactone facilitates the early development of parthenogenetically activated porcine embryos by reducing oxidative stress and inhibiting autophagy

Wang Xin-Qin 1
Liu Rong-Ping 1
Wang Jing 1
Luo Dan 2
Li Ying-Hua 1
http://orcid.org/0000-0003-2008-6166 Jiang Hao 2
Xu Yong-Nan 1 ynxu0613@163.com
Kim Nam-Hyung 1 namhyungkim@163.com
1 Guangdong Provincial Key Laboratory of Large Animal Models for Biomedicine , School of Biotechnology and Health Sciences, Wuyi University, Jiangmen , China
2 Jilin Provincial Key Laboratory of Animal Model, Jilin University , Changchun, Jilin , China
Mokhtar Mohd Helmy
Electronic publication date: 2022 Jul 25
Publication date: 2022
Volume: 10
Electronic Location ID: e13766
Received 2022 Mar 16; Accepted 2022 Jun 30
Copyright: © 2022 Wang et al.
Copyright year: 2022
Copyright holder: Wang et al.
License: This is an open access article distributed under the terms of the Creative Commons Attribution License, which permits unrestricted use, distribution, reproduction and adaptation in any medium and for any purpose provided that it is properly attributed. For attribution, the original author(s), title, publication source (PeerJ) and either DOI or URL of the article must be cited.
License URL: https://creativecommons.org/licenses/by/4.0/

Keywords: Wedelolactone, Porcine embryo, Oxidative stress, Autophagy, Nrf2/ARE

Funding: Guangdong Provincial Department of Science and Technology 2021B1212040016 Guangdong Provincial Department of Education 2021ZDZX2046 This study was supported by the science and technology planning project of Guangdong Provincial Department of Science and Technology (Project No.: 2021B1212040016); and the special project in key areas of biomedicine and health of Guangdong Provincial Department of Education (Project No.: 2021ZDZX2046). The funders had no role in study design, data collection and analysis, decision to publish, or preparation of the manuscript.

==============================
Wedelolactone (WDL) is a coumaryl ether compound extracted from the traditional Chinese medicinal plant, Eclipta prostrata L. It is a natural polyphenol that exhibits a variety of pharmacological activities, such as anti-inflammatory, anti-free radical, and antioxidant activities in the bone, brain, and ovary. However, its effect on embryonic development remains unknown. The present study explored the influence of WDL supplementation of porcine oocytes culture in vitro on embryonic development and the underlying mechanisms and its effect on the levels of Kelch-like ECH-associated protein 1/nuclear factor-erythroid 2-related factor 2/antioxidant response element (Keap1/Nrf2/ARE). The results showed that WDL (2.5 nM) significantly increased the blastocyst formation rate, mitochondrial activity, and proliferation ability while reducing the reactive oxygen species accumulation, apoptosis, and autophagy. These findings suggested that WDL can enhance the growth and development of early porcine embryos to alleviate oxidative stress and autophagy through regulating NRF2 and microtubule-associated protein 1 light chain 3 (MAP1LC3) gene expression levels.

Introduction

Fertilized eggs undergo mitosis to form 2-, 4-, and 8-cell embryos, morula, and establish the first internal cavity to generate a hollow spherical blastocyst, which is finally implanted in the uterus (Zhu & Zernicka-Goetz, 2020). The transition from fertilized eggs to blastocyst before implantation is called early embryonic development (Hoppe & Illmensee, 1982). Compared with in vivo conditions, embryo culture in vitro is affected by many environmental factors, such as osmotic pressure, temperature, and gas composition (Watkins et al., 2007). With an increase in oxygen content in vitro, the embryo lacks protection from the natural antioxidant system, which leads to a continuous increase in oxidative stress (Karagenç et al., 2019). Furthermore, excess reactive oxygen species (ROS) production and excessive autophagy can affect various redox reactions in the energy production process, inhibiting the expression of endogenous antioxidant molecules, such as reduced glutathione (GSH), and various antioxidant enzymes, thereby impairing the mitochondrial function and DNA expression and inhibiting the embryonic development (Coffman & Su, 2019).

Wedelolactone (WDL), 1,8,9-Trihydroxy-3-methoxy-6H-[1]benzofuro[3,2-c][1]benzopyran-6-one (C16H10O7), is a natural coumaryl ether compound isolated from the traditional Chinese medicinal plant, Wedelia chinensis (Eclipta prostrata L.) (Hsieh et al., 2015). WDL has many pharmacological activities, such as hepatoprotection, immunomodulatory, lipid metabolism, antitumor, antihypoxic, anti-free radical, and antioxidant functions (Benes et al., 2012; Jayathirtha & Mishra, 2004; Kim et al., 2017; Zhao et al., 2015). Administration with WDL (10 and 20 μM) mitigated acute pancreatitis and associated lung injury by reducing the levels of serum pancreatic digestive enzymes and pro-inflammatory cytokines (Fan et al., 2021). Furthermore, supplementation of WDL (37.5 μM) alleviated various stress responses, mitochondrial function, lipid levels, and α-syn deposition to mitigate Parkinson’s disease via the nuclear factor-erythroid 2-related factor 2 (Nrf2)/skinhead-1 (SKN-1) signaling pathway (Sharma et al., 2021). However, the effect of WDL on porcine early preimplantation embryo development remains unclear.

The Kelch-like ECH-associated protein 1 (Keap1)/Nrf2/antioxidant response element (ARE) pathway is the central defense mechanism against oxidative stress (Hassanein et al., 2020). Nrf2 is a transcription factor that regulates the cellular defense against ROS and chemical assaults by modulating the expression of genes involved in chemical detoxification and the oxidative stress response (Wang et al., 2019). Keap1 is a multiregional inhibitory protein that mainly exists in the cytoplasm (Ortet et al., 2021). It is a cytoplasmic inhibitor protein of Nrf2. It is usually attached to the actin cytoskeleton of the cytoplasm and mediates Nrf2 ubiquitination and subsequent proteasomal degradation by acting as a linker molecule for the cullin-RING E3 ligase, CUL-E3 (Chen, 2022; Raghunath et al., 2018). Nrf2 enters the nucleus, separates from Keap1, and combines with ARE to promote the transcription of antioxidant genes, thereby inhibiting oxidative stress (Villeneuve, Lau & Zhang, 2010; Xu et al., 2016).

Previous research has shown that melatonin treatment of in vitro fertilization-derived porcine embryos prevents ROS generation via the Nrf2/ARE signaling pathway (Kim et al., 2019). Alpha-ketoglutarate improved the meiotic maturation of porcine oocytes and the quality of parthenogenesis embryos by reducing oxidative stress through activation of the Nrf2 pathway and, subsequently, its downstream cellular antioxidant response mechanisms (Chen et al., 2022). Sulforaphane protected human granulosa cells against hydrogen peroxide (H2O2)-induced oxidative stress by reducing the intracellular ROS production and subsequent apoptosis through an Nrf2-mediated increase in the antioxidant enzymes (Esfandyari et al., 2021). However, whether WDL can improve oxidative stress in early preimplantation porcine embryos by regulating the Nrf2/ARE pathway remains unclear. Therefore, the purpose of this study was to investigate whether WDL can promote the early development of parthenogenetically activated (PA) porcine embryos in vitro and to explore its related mechanism, such as antioxidation and inhibition of autophagy.

Materials and Methods

Animals and chemicals

Unless stated otherwise, all chemicals and reagents were purchased from Sigma-Aldrich (St. Louis, MO, USA). Porcine ovaries used in this study were obtained from a slaughterhouse (Jiangmen, China). They were not required to be screened by the Institutional Animal Care and Use Committee (IACUC) because our university does not regard the use of these ovaries as an animal experiment. WDL (Selleck Chemicals, Houston, TX, USA) was dissolved in dimethyl sulfoxide (DMSO) to obtain a concentration of 25 µM and further diluted to 0.25, 2.5, and 25 nM for the experiment. For exact comparison, porcine zygote medium-5 (PZM-5) was treated with each concentration.

Oocyte collection and in vitro maturation (IVM)

Prepubertal gilt ovaries were obtained from a local abattoir and transported to the laboratory in a 0.9% sterile saline solution supplemented with penicillin G and streptomycin sulfate. The cumulus-oocyte complexes (COCs) were aspirated from 3–8 mm follicles in the ovary with an 18-gauge needle fitted on a disposable 10 mL syringe (Sánchez et al., 2017). The suction follicle solution was dispensed into a 15 mL glass tube and precipitated. Then, the supernatant was removed and washed thrice in Tyrode’s lactate HEPES medium and transferred to 0.5 mL of IVM medium (M199 medium with 0.1 mL of porcine follicular fluid, 0.91 mM sodium pyruvate, 10 ng/mL of epidermal growth factor, 1 mg/mL insulin, 10 IU/mL of follicle-stimulating hormone, and 10 IU/mL of luteinizing hormone) covered with mineral oil in a 4-well culture plate and incubated at 38.5 °C in an atmosphere of 5% CO2 and 100% relative humidity (RH) for 44–46 h.

Parthenogenetic activation and in vitro culture (IVC)

In brief, after IVM, oocytes and cumulus cells were separated from COCs by pipetting approximately 30 times with 0.1% hyaluronidase in a 1.5 mL centrifuge tube (Van de Velde et al., 1997). The cumulus-depleted oocytes were subjected to parthenogenetic activation using two pulses of direct current at 120 V for 60 µs in 300 mM mannitol containing 0.5 mM HEPES, 0.05 mM CaCl2•2H2O, 0.1 mM MgSO4•7H2O, and 0.01% polyvinyl alcohol (PVA) and cultured in 7.5 mg/mL cytochalasin B at 38.5 °C in an atmosphere of 5% CO2 and 100% RH for 3 h (Ock et al., 2011). Subsequently, approximately 40 oocytes were washed thoroughly with the IVC medium, transferred to 500 mL of IVC medium (bicarbonate-buffered PZM-5 supplemented with 4 mg/mL of bovine serum albumin (BSA)) covered with mineral oil in a 4-well culture plate, and incubated at 38.5 °C in an atmosphere of 5% CO2 and 100% RH for 7 days. WDL was first dissolved in DMSO to prepare a 100 mM stock solution. Then, it was dissolved in ddH2O to achieve a concentration of 1 mM. We determined the dose-dependent effects of different concentrations (0, 0.25, 2.5, and 25 nM) of WDL on early porcine embryos cultured in vitro.

Terminal deoxynucleotidyl transferase-mediated 2′-deoxyuridine 5′-triphosphate (dUTP) nick-end labeling (TUNEL) assay

Apoptosis was analyzed using a TUNEL Assay Kit (Roche Diagnostics, Indianapolis, IN, USA) according to the manufacturer’s instructions. Briefly, blastocysts on day 7 were washed four times with 0.1% polyvinyl alcohol in phosphate-buffered saline (PBS-PVA) and fixed with 3.7% paraformaldehyde in PBS-PVA for 30 min. Then, the embryos were permeabilized by an additional incubation in 0.1% Triton X-100 at room temperature (RT) for 40 min. Next, the embryos were incubated with 10% terminal deoxynucleotidyl transferase enzyme and fluorescein-conjugated dUTP (Roche Diagnostics, Indianapolis, IN, USA) at RT for 1 h in the dark. Subsequently, the embryos were incubated with 10 mg/mL of Hoechst 33342 at RT for 10 min to label the nuclei. Finally, the embryos were washed four times with PBS-PVA and mounted on a glass slide. An inverted fluorescence microscope (Ti2eU; Nikon, Tokyo, Japan) and ImageJ software (NIH, Bethesda, MD, USA) were used to analyze the number of fluorescent cells. The apoptosis rate was calculated as the ratio of TUNEL-positive cells to the total number of cells.

5-Ethynyl-2′-deoxyuridine (EdU) assay

Cell proliferation was measured using the BeyoClick EdU-555 Cell Proliferation Assay Kit (Beyotime, Shanghai, China) according to the manufacturer’s instructions. In short, on day 6, the embryos were transferred to fresh IVC medium containing 10% EdU with or without WDL and then incubated at 38.5 °C in an atmosphere of 5% CO2 and 100% RH for 8 h. Next, the blastocysts were fixed with 3.7% paraformaldehyde in PBS-PVA for 30 min, and then permeabilized in 0.1% Triton X-100 at RT for 30 min. After washing four times with PBS-PVA, the embryos were incubated with 5% BeyoClick Additive Solution at 38.5 °C in an atmosphere of 5% CO2 and 100% RH for 14 h. Then, the embryos were incubated with 10 mg/mL of Hoechst 33342 at RT for 10 min to label the nuclei. Finally, the embryos were washed four times with PBS-PVA and mounted on glass slides. An inverted fluorescence microscope (Ti2eU; Nikon, Tokyo, Japan) and ImageJ software (NIH) were used to count the number of EdU-positive cells and the total cells. The proliferation rate was calculated as the ratio of EdU-positive cells to the total number of cells.

Intracellular ROS and glutathione (GSH) level assay

To measure the intracellular ROS and GSH levels, 4-cell-stage embryos were treated in 1 mL of PBS-PVA medium containing 10 mM 2′,7′-dichlorodihydrofluorescein diacetate (H2DCFDA; Invitrogen, Rochester, NY, USA) and 10 mM 4-chloromethyl-6,8-difluoro-7-hydroxycoumarin (CMF2HC; Invitrogen, Rochester, NY, USA) at 38.5 °C in an atmosphere of 5% CO2 and 100% RH for 1 h. After washing four times with PBS-PVA, the stained embryos were put into a 5 µL droplet of PBS-PVA, photographed with an inverted fluorescence microscope (Ti2eU; Nikon, Tokyo, Japan), and the fluorescence intensities of the cells were analyzed with ImageJ software (NIH).

Mitochondrial membrane potential (MMP, Δψm) assay

To measure the MMP (ΔΨm), 4-cell-stage embryos were incubated in Medium 199 containing 10 µg/mL 5,50,6,60-tetrachloro-1,1′,3,3′-tetraethylbenzi-midazolylcarbocyanine iodide (JC-1, Beyotime, Shanghai, China) dye at 38.5 °C in an atmosphere of 5% CO2 and 100% RH for 1 h. After washing with PBS-PVA four times, the stained embryos were put into a 5 µL droplet of PBS-PVA, and images were captured using a camera attached to an inverted fluorescence microscope (Ti2eU; Nikon, Tokyo, Japan). Fluorescence intensities of the red and green fluorescence signals were analyzed using the ImageJ software (NIH). The average MMP levels of all 4-cell-stage embryos were calculated as the ratio of the red fluorescence intensity (J-aggregate) to the green fluorescence intensity (J-monomer).

Immunofluorescence staining and confocal microscopy

The day-7 embryos were collected and fixed with 3.7% paraformaldehyde in PBS-PVA for 30 min, and then permeabilized with PBS-PVA containing 0.1% Triton X-100 at RT for 30 min. Afterward, the embryos were incubated in PBS-PVA containing 3% BSA at RT for 1 h and then incubated with primary antibody LC3B antibody (#ab48394, diluted 1:200; Abcam, Cambridge, MA, USA) and rabbit polyclonal anti-NRF2 antibody (#ab31163, diluted 1:200; Abcam, Cambridge, MA, USA) at 4 °C for 14 h. After washing thrice with PBS-PVA, the cells were incubated with goat anti-rabbit IgG (#ab150077, diluted 1:500; Abcam, Cambridge, MA, USA) for 1 h, followed by incubation with 10 mg/mL of Hoechst 33342 at RT for 10 min to label the cell nuclei. Finally, the embryos were washed four times with PBS-PVA and mounted on glass slides. The number of microtubule-associated protein 1 light chain 3 (MAP1LC3) spots and the fluorescence intensity of Nrf2 were detected using an inverted fluorescence microscope (Ti2eU; Nikon, Tokyo, Japan), and the fluorescence intensity was analyzed with ImageJ software (NIH).

Quantitative real-time reverse transcription-polymerase chain reaction (qRT-PCR)

Ten day-7 embryos of similar size were selected from the control and treated groups, respectively. Total mRNA was extracted using the Dynabeads mRNA DIRECT Purification Kit (Invitrogen, Rochester, NY, USA), according to the manufacturer’s instructions. Reverse transcription to first-strand cDNA was performed using a reverse transcription kit (Tiangen Biotech, Beijing, China). Each 20 µL of qRT-PCR reaction mixture consisted of 10 µL of KAPA SYBR FAST qPCR Master Mix (2X) Universal, 8.8 µL of template DNA (<20 ng/20 µL), 0.4 µL of ROX Low, 0.4 µL of forward primer, and 0.4 µL of reverse primer (10 µM, Table 1). The qRT-PCR conditions included denaturation at 95 °C for 30 s, followed by 40 cycles at 95 °C for 5 s, 60 °C for 15 s, and 72 °C for 30 s. Gene expression was quantified using the LightCycler96 and the 2−ΔΔCt method based on 18S rRNA.

Table 1 Information about the primer sequences.

Genes	Primer sequences (5′–3′)	Anneal temperature (°C)	
Forward	Reverse	
NANOG	AAGTACCTCAGCCTCCAGCA	GTGCTGAGCCCTTCTGAATC	60	
OCT4	GTGAGAGGCAACCTGGAGAG	TCGTTGCGAATAGTCACTGC	60	
CASP3	GACGGACAGTGGGACTGAAGA	GCCAGGAATAGTAACCAGGTGC	60	
SIRT1	ATCGTCACCAATGGTTTCCA	GGATCTGTGCCAATCATGAG	60	
SOX2	AAGAGAACCCCAAGATGCACAACT	GCTTGGCCTCGTCGATGAAC	60	
ATG5	GGTTTGAATATGAAGGCACACCA	TGTGATGTTCCAAGGAAGAGCTG	60	
LC3	AACGAAATTCCTGGTGCCTGA	AAGGCTTGGTTAGCATTGAGCTG	60	
P62	AAGAGAAGCCGCCTGACAC	CGACTCCAGGGCGATCTTAT	60	
KEAP1	ACCCAATTTCTGCCCCTGAG	ACTTGACCTGCAGCGTAACA	60	
NRF2	GCCCAGTCTTCATTGCTCCT	AGTCCTCCCAAACTTGCTC	60	
UCHL1	ACTTTGGATTCGCTCGGTAC	CGCTTATCTGCAGACCCCAA	60	
GPX1	CACAACGGTGCGGGACTA	CATTGCGACACACTGGAGAC	60	
NOX2	TGTATCTGTGTGAGAGGCTGGTG	CGGGACGCTTGACGAAA	60	
SOD1	TGACTGCTGGCAAAGATGGT	TTTCCACCTCTGCCCAAGTC	60	
GAPDH	TTCCACGGCACAGTCAAG	ATACTCAGCACCAGCATCG	60	

Statistical analysis

The statistical results are expressed as the mean ± standard deviation (SD). A Student’s t-test was used to compare the data of the two groups. One-way analysis of variance (ANOVA, Tukey-Kramer test) was used to analyze three or more mean values. All statistical analyses were performed at *P < 0.05, **P < 0.01, and ***P < 0.001 using SPSS version 22.0 (IBM Corp., Chicago, IL, USA).

Results

Effects of different concentrations of WDL on the early developmental rates of porcine embryos

We examined the blastocyst formation rates on days 6 and 7 following exposure to different concentrations of WDL (Fig. 1A). The results showed that 2.5 nM WDL increased the blastocyst formation rate on day 6 (34.73 ± 1.02% vs. 27.27 ± 0.29%, 26.13 ± 0.59%, and 28.60 ± 0.31%; **P < 0.01; Fig. 1B) and day 7 (45.40 ± 1.06% vs. 35.47 ± 1.34%, 32.27 ± 1.19%, and 35.73 ± 0.37%; **P < 0.01, Fig. 1B) compared to the control, 0.25, and 25 nM WDL. The total blastocyst cell numbers in the 0 and 2.5 nM WDL-treated groups were 34.63 ± 0.97 and 40.89 ± 1.03, respectively (**P < 0.01, Fig. 1C). Thus, subsequent experiments were performed using 2.5 nM WDL.

Figure 1 Effects of different concentrations (0, 0.25, 2.5, or 25 nM) of wedelolactone (WDL) on porcine blastocyst formation rate on day 6 and day 7.

(A) Embryonic development rate from day 6 to day 7 in the control and WDL-treated groups. Scale bars represent 100 μm. (B) Blastocyst formation rates on day 6 and day 7. Significant differences are represented by two asterisks (**) indicating P < 0.01. (C) Total number of cells in blastocysts on day 7 with (n = 90) or without WDL treatment (n = 86). Data are presented as the mean ± standard deviation (SD). Significant differences are represented by two asterisks (**) indicating P < 0.01.

WDL reduced apoptosis and improved cell proliferation in blastocysts

To explore why 2.5 nM WDL treatment improved the quality of porcine PA embryos, we first detected the cell proliferation ability and apoptosis in blastocysts with or without WDL treatment (Pang et al., 2013). The number of EdU-positive nuclei in the WDL-treated PA embryos was increased by 1.25 ± 0.09-fold compared to the control group (**P < 0.01, Figs. 2A and 2C). The ratio of TUNEL-positive nuclei in the WDL-treated PA embryos was lower (4.43 ± 0.18%; ***P < 0.001) than that in the control group (7.33 ± 0.31%; Figs. 2B and 2D). Expectedly, the mRNA expression levels of NANOG, organic cation transporter 4 or octamer-binding transcription factor 4 (OCT4), and sex-determining region Y (SRY)-box transcription factor 2 (SOX2) were significantly increased in the 2.5 nM WDL group, and the mRNA expression levels of caspase 3 (CASP3) and sirtuin 1 (SIRT1), the apoptosis-related genes, were significantly reduced in the WDL-treated groups compared to the control (**P < 0.01, Fig. 2E).

Figure 2 Effects of wedelolactone (WDL) on blastocyst cell proliferation and apoptosis.

(A) Representative 5-ethynyl-2′-deoxyuridine (EdU) staining images of blastocysts. Scale bars represent 200 μm. (B) Representative terminal deoxynucleotidyl transferase-mediated 2′-deoxyuridine 5′-triphosphate (dUTP) nick-end labeling (TUNEL) staining images of blastocysts. (C) Relative levels of proliferating cells in embryos treated with (n = 52) or without WDL (n = 41) on day 7. Significant differences are represented by three asterisks (***) indicating P < 0.001. (D) Proportions of apoptotic cells in embryos treated with (n = 102) or without (n = 96) WDL on day 7. Significant differences are represented by three asterisks (***) indicating P < 0.001. (E) Effect of WDL on expression levels of blastocyst multipotency and apoptosis-related genes.

WDL protects against ROS and improves GSH levels

To explore the ability of WDL to protect the embryonic cells from oxidation and its role in improving embryonic development, we treated the 4-cell stage porcine embryos with H2DCFDA and CMF2HC for detection of ROS and GSH, respectively (Yao et al., 2019). Quantitative analysis of the fluorescence intensity indicated that WDL-treated embryos showed a significant decrease in total ROS compared with control embryos (1.36 ± 0.04-fold, ***P < 0.001, Figs. 3A and 3B), while cotreatment with WDL led to an increase in the fluorescence intensity of CMF2HC (1.37 ± 0.05-fold, ***P < 0.001, Figs. 3C and 3D), illustrating that the cell viability was significantly higher in the WDL-treated group than in the control group. There was a substantial increase in the mRNA expression levels of oxidative stress-related genes, glutathione peroxidase 1 (GPX1), NADPH oxidase 2 (NOX2), and superoxide dismutase 1 (SOD1) in WDL-treated embryos compared with that in the control embryos (P < 0.001, Fig. 3E).

Figure 3 Effects of wedelolactone (WDL) on the levels of reactive oxygen species (ROS) and glutathione (GSH) in porcine embryos.

(A) Representative 2′,7′-dichlorodihydrofluorescein diacetate (H2DCFDA) staining images of 4-cell stage embryos. Scale bars represent 100 μm. (B) Relative H2DCFDA fluorescence intensity level changes in 4-cell stage embryos in the control (n = 83) and WDL-treated (n = 81) groups. Significant differences are represented by three asterisks (***) indicating P < 0.001. (C) Representative 4-chloromethyl-6, 8-difluoro-7-hydroxycoumarin (CMF2HC) staining images of 4-cell stage embryos. Scale bars represent 100 μm. (D) Relative CMF2HC fluorescence intensity changes of 4-cell stage embryos in control (n = 89) and WDL-treated (n =92) groups. Significant differences are represented by three asterisks (***) indicating P < 0.001. (E) Effect of WDL on expression levels of blastocyst oxidative stress-related genes (three asterisks (***) indicating P < 0.001).

WDL reduces autophagy during early-stage porcine embryo development

Autophagy is the main pathway leading to the sequestration of cytoplasmic proteins/molecules by lysosomes (Baba et al., 1997). Oxidative stress and mitochondrial dysfunction can also induce cell autophagy (Sachdeva et al., 2019). The MAP1LC3 protein is widely used as a marker for autophagy. Therefore, to evaluate whether WDL can affect autophagy, we examined the levels of MAP1LC3 in WDL-treated groups. MAP1LC3 immunofluorescence staining revealed that the number of MAP1LC3 dots in blastocysts treated with WDL was significantly reduced in the same section compared to the control group (0.76 ± 0.12-fold, **P < 0.01, Figs. 4A and 4B) and that WDL administration also decreased the mRNA gene expression levels of autophagy-related gene 5 (ATG5), MAP1LC3, and BECLIN1 compared to the control group (*P < 0.05, Fig. 4C). These results showed that WDL reduced autophagy.

Figure 4 Effect of wedelolactone (WDL) on autophagy expression in porcine blastocysts.

(A) Representative fluorescence images show the presence of the light chain 3 (LC3) protein in porcine blastocysts treated with (n = 46) or without WDL (n = 34). Scale bars represent 200 μm. (B) Fluorescence dots denote the number of LC3 in blastocysts. Significant differences are represented by two asterisks (**) indicating P < 0.01. (C) Effect of WDL on expression levels of blastocyst autophagy-related genes.

WDL enhances mitochondrial function during early-stage porcine embryo development

The MMP (ΔΨm) is often used as an indicator of mitochondrial function and is an important factor affecting early embryo development (Zhou et al., 2016). The MMP was evaluated in porcine embryos by the JC-1 reaction. The results showed that the ratio of fluorescence intensity (red/green) was significantly higher in the WDL treatment group than in the control group (1.17 ± 0.05-fold, ***P < 0.001, Figs. 5A and 5B). These results suggested that WDL may upregulate mitochondrial function to protect against mitochondrial dysfunction and deficiency.

Figure 5 Effect of wedelolactone (WDL) on mitochondrial function in 4-cell stage embryos.

(A) Representative fluorescent images of JC-1-stained 4-cell stage embryos in control and WDL-treated groups. Scale bars represent 100 μm. (B) Relative fluorescence levels of JC-1 red/green in 4-cell stage embryos treated with (n = 67) or without (n = 60) WDL. Significant differences are represented by three asterisks (***) indicating P < 0.001.

Effect of WDL treatment during IVC on Nrf2/ARE in porcine blastocysts

Immunofluorescence staining of Nrf2 in blastocysts treated with WDL was increased compared to the control group (1.27 ± 0.12-fold, **P < 0.01, Figs. 6A and 6B). Afterward, the expression levels of genes related to the Nrf2/ARE signaling pathway (ubiquitin C-terminal hydrolase L1 (UCHL1), heme oxygenase-1 (HO-1), quinone oxidoreductase (NQO1), KEAP1) were evaluated by qRT-PCR. WDL significantly upregulated the Nrf2 gene expression level and ARE compared to the control group (*P < 0.05 and **P < 0.001, respectively; Fig. 6C). However, the mRNA expression of KEAP1 was not affected by treatment with WDL.

Figure 6 Effect of wedelolactone (WDL) on the expression levels of the nuclear factor-erythroid 2-related factor 2 (Nrf2)/antioxidant response element (ARE) signaling pathway-related genes in porcine blastocysts.

(A) Representative porcine blastocysts were stained with Nrf2 and counterstained with Hoechst 33342 in control (n = 60) and WDL-treated groups (n = 79). Scale bars represent 200 μm. (B) The experiment was replicated at least three times. Fluorescence intensity of Nrf2 in blastocysts. Significant differences are represented by two asterisks (**) indicating P < 0.01. (C) Nrf2/ARE signaling pathway-related genes in blastocysts.

Discussion

Although the development of IVC technology is relatively mature, embryos cultured in an IVC environment have poorer developmental ability than embryos cultured in vivo, which lack the protection afforded by the maternal endocrine environment (Niu et al., 2019; Tam, 2019). In the present study, we investigated the effect of the antioxidant WDL on porcine embryonic development after parthenogenetic activation. Previous research has shown that WDL can inhibit breast cancer cell growth and promote bone marrow mesenchymal stem cell proliferation by inhibiting nuclear factor-kappa B (NF-κB)a transcription factorand/or androgen receptor (Deng et al., 2019; Nehybová et al., 2017). Therefore, we explored whether WDL could improve blastocyst quality by reducing oxidative stress levels and affecting related pathways.

During the 6–7 days following fertilization, the zygote undergoes multiple rounds of division to form an outer trophectoderm layer surrounding an inner cell mass and a fluid-filled cavity (Popovic, Azpiroz & Chuva de Sousa Lopes, 2021). Consequently, we chose the day-7 blastocyst for the index comparison test to better evaluate the impact of WDL on embryonic developmental potential. The average binding rate of WDL to plasma proteins is high (>80%), which means that the drug mainly exists in a bound form and has a long plasma half-life. Hence, within 7 days of adding WDL to the IVC culture in this experiment, there was no need to change the medium or add fresh drugs. Conversely, compared with the optimal concentration of other natural medicines for embryonic development, the optimum concentration of WDL was found to be low (2.5 nM). For example, 40 μM imperatorin can improve porcine preimplantation embryonic development in vitro by reducing oxidative stress and autophagy (Luo et al., 2020). However, according to the related experiments on WDL, a low dose of 2 µg/mL (6.4 nM) WDL promoted the osteogenesis of bone marrow mesenchymal stem cells (Deng et al., 2019; Zhu et al., 2018).

A recent study showed that ecto-5′-nucleotidase (CD73) could promote the proliferation of breast cancer cells and breast cells in vitro through the AKT/glycogen synthase kinase-3β (GSK-3β)/β-catenin/cyclin D1 signaling pathway. A high concentration of WDL (100 μM) inhibited the expression of CD73 by inhibiting this pathway, thereby promoting the accumulation of cisplatin in ovarian cancer cells and improving the treatment efficacy of drug-resistant ovarian cancer cells (Sarwar et al., 2021). According to our experimental results, high concentrations of WDL inhibit development, while low concentrations promote early embryonic development. Consequently, the inhibition of embryonic development at high concentrations may be because WDL inhibits β-catenin protein expression, which affects subsequent histone acetylation, thereby impairing various processes, such as cell cycle progression, gene transcription, and DNA damage repair.

In endogenous metabolic reactions, aerobic cells produce ROS, such as superoxide anion (O2−), H2O2, hydroxyl radicals (OH•), and organic peroxides, as normal molecular oxygen products (Nosaka & Nosaka, 2017). Therefore, we predicted that WDL would potentially benefit the development of preimplantation embryos and fetuses as a substance with antioxidant capacity. In this study, WDL effectively alleviated oxidative stress, stabilized mitochondrial function, increased cell proliferation, and reduced apoptosis and autophagy. This may have been because WDL regulates the levels of ROS, which can trigger uncontrolled reactions with non-target intracellular compounds, oxidizing nucleic acids, proteins, cell membranes, and other lipids (Eckl & Bresgen, 2017). Among several important targets of oxidative attack are SOD, GPX, and NADH oxidase (Junod, 1989). As expected, our results were accompanied by up-regulation of SOD1 and GPX1. Another explanation for our observations is that WDL promoted the expression of pluripotency genes in cells, and the pluripotency factors NANOG and OCT4 continued to promote cell proliferation (Deng et al., 2019). For example, endogenous NANOG can drive embryonic stem cell self-renewal, whereas elevated Nanog expression from transgene constructs maintains OCT4 levels sufficient for human embryonic stem cell proliferation (Chambers et al., 2003). Therefore, WDL can stimulate the expression of antioxidant-related genes, reduce apoptosis, and promote cell proliferation to overcome the detrimental effects of harmful microenvironments on the embryo.

Finally, we investigated the effect of WDL on the activation of the Nrf2/ARE signaling pathway in blastocysts. ARE is the core protective sequence of the promoter region of the cytoprotective protein gene (Wasserman & Fahl, 1997). The 5′-promoter region of many phase II metabolic enzyme genes contains ARE, which can be subject to oxidation and pre-activation by electrochemical complexes (Nakagami, 2016). Oxidative stress induces Nrf2-Keap1 dissociation and Nrf2 nuclear translocation. In the nucleus, Nrf2 forms a heterodimer with the musculoaponeurotic fibrosarcoma (MAF) transcription factor. Subsequently, Nrf2/MAF can bind ARE and transcribe downstream genes. Common downstream gene products include HO-1, NAD(P)H dehydrogenase, NQO1, glutamylcysteine ligase, c-glutamylcysteine synthase, GPX1, glutathione S-transferase, glutathione reductase, HO-1, and SOD (Su et al., 2013). In accordance with our hypothesis, WDL supplementation improved porcine embryo IVC and increased the mRNA transcript levels of Nrf2/ARE signaling-related genes (Keap1, UCHL1, SOD1, and HO-1).

During the rapid development of the preimplantation embryo, the embryo undergoes extensive cell proliferation and cell turnover, accompanied by a high amount of programmed cell death (Lockshin & Zakeri, 2004). Cell proliferation and cell death ensure normal cell growth by maintaining a dynamic balance of relevant indices, such as maintaining normal embryonic development through phagocytosis of damaged or dead cell bodies when the embryonic genome is activated (Jukam, Shariati & Skotheim, 2017). However, when cells are unable to balance the damage mediated by oxidative stress, excessive autophagy results in autophagic cell death. For example, exposure of porcine embryos to high concentrations of H2O2 induces microtubule-associated protein 1 light chain 3 beta (MAP1LC3B) and lysosome-associated membrane protein-2 (LAMP2) gene expression and MAP1LC3 accumulation (Xu et al., 2011). In our study, the expression levels of autophagy-related genes (ATG5, LC3, and BECLIN1) were downregulated by WDL supplementation at the blastocyst stage, indicating that autophagy genes interact with the Nrf2/ARE system to maintain redox homeostasis and protect the embryos from adaptive stress. However, due to the differences between PA and fertilized embryos, further studies are needed to validate the role of WDL in embryos produced via in vitro fertilization and intracytoplasmic sperm injection, explore its effects on Nrf2-related factors, such as NF-κB and p62, and to elucidate its roles in gamete maturation and early embryonic development.

Conclusion

Our results indicate that WDL (2.5 nM) supplementation enhances the growth and development of early porcine embryos by decreasing oxidative stress, apoptosis, and autophagy, while increasing pluripotency, mitochondrial function, and cell proliferation. Additionally, WDL (2.5 nM) supplementation also increases the anti-oxidase levels to prevent oxidative stress in early embryos by regulating Nrf2/ARE and autophagy-related gene expression levels to support further optimization of embryo IVC systems.

Supplemental Information

Supplemental Information 1 Data on blastocyst rate, cell number, proliferation rate, apoptosis rate, and mRNA expression of apoptotic and proliferation-related genes.

Click here for additional data file.

Supplemental Information 2 Data on fluorescence expression of ROS, GSH, LC3, JC-1, NRF2, and mRNA expression of antioxidant and autophagy-related genes.

Click here for additional data file.

Additional Information and Declarations

Competing Interests

Author Contributions

Data Availability

The authors declare that they have no competing interests.

Xin-Qin Wang conceived and designed the experiments, performed the experiments, analyzed the data, prepared figures and/or tables, and approved the final draft.

Rong-Ping Liu performed the experiments, prepared figures and/or tables, investigation, and approved the final draft.

Jing Wang performed the experiments, prepared figures and/or tables, investigation, and approved the final draft.

Dan Luo performed the experiments, prepared figures and/or tables, conceptualization, and approved the final draft.

Ying-Hua Li performed the experiments, authored or reviewed drafts of the article, methodology, and approved the final draft.

Hao Jiang conceived and designed the experiments, authored or reviewed drafts of the article, and approved the final draft.

Yong-Nan Xu conceived and designed the experiments, authored or reviewed drafts of the article, and approved the final draft.

Nam-Hyung Kim conceived and designed the experiments, analyzed the data, authored or reviewed drafts of the article, and approved the final draft.

The following information was supplied regarding data availability:

Data is available at ZENODO (DOI 10.5281/zenodo.6513530) and raw measurements are available in the Supplemental Files.

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
