# Peer review of "Wedelolactone facilitates the early development of parthenogenetically activated porcine embryos by reducing oxidative stress and inhibiting autophagy"

_PeerJ, doi:10.7717/peerj.13766_

## Round 0.1 · original submission · Major Revisions

Thank you for the submission. In reviewing the information provided by the reviewers, I am requesting significant revisions before the paper can be considered further.

Reviewer 1 ·

Basic reporting

1. The article must be written in English and must use clear, unambiguous, technically correct text. The article must conform to professional standards of courtesy and expression.
2. The title should be change to the interesting topic.
3. Abstract
- What is a couma ether compound extracted? (no mention in the manuscript)
- "in vitro" this word should change into italic letter in all of manuscript
- In abstract should be write the conclusion in the last sentence.
4. Introduction :
- more review about Nrf2/ARE signaling pathway in embryo development
- No research aims and scope

Experimental design

1. No subtopic of experimental design. The submission should clearly define the research question, which must be relevant and meaningful. The knowledge gap being investigated should be identified, and statements should be made as to how the study contributes to filling that gap.
2. line 59-60 move to combine with subtopic "Animals and chemicals"
3. Why some assay was done on day 6 and day 7 of embryo, should explain?

Validity of the findings

1. line 173-175 : should rewrite because redundant writing in 2 sentences.
2. line 189 : what is NC group? no mention before
3. line 249-250 : the unit of measure was difference, how to compare it?
4. line 253-255 : should give more reason the support this data

Additional comments

Please double check the correctness of references.

Reviewer 2 ·

Basic reporting

The language of this paper is professional, the background introduction is acceptable, the structure of the article is clear, the charts are used properly, and the results are complete.

Experimental design

In this paper, the improvement of natural compound wedelolactone (WDL) on animal early embryonic development was reported for the first time, and the related mechanism was explored, which has reference value for further optimizing embryo culture medium. A variety of conventional embryo culture and detection techniques were used in this study, but some descriptions in this paper are not specific and clear enough and need to be supplemented and improved.

Validity of the findings

This paper found that WDL can improve the in vitro development rate of porcine parthenogenetic early embryos, improve mitochondrial function, reduce ROS, and reduce the level of apoptosis and autophagy. However, the research on relevant molecular mechanisms is relatively weak, some results are unclear and inaccurate, and there is a lack of difference statistical results. Discussion in view of some results is unreasonable and imperfect, and it is necessary to further consult the literature to supplement the discussion.

Additional comments

Major points
1. Please describe the duration of WDL treatment, how long the half-life of WDL is and how the control group is treated. Whether it is necessary to change the medium and add fresh WDL during embryo culture.

2. How many embryos were used for blastocyst staining and difference statistics? Please specify.

3. Line105-106 does not need to count the fluorescence density, but the number of bright spots; Line 150-151 need to count the fluorescence intensity (Nrf2) and the number of bright spots (LC3) respectively. How to count the number of bright spots in blastocysts?

4. Fig2b. TUNEL color is not clear, and the merge diagram should be given. C and D should be compared with the total number of positive cells (expressed in %), as described in line105-106.

5. Fig4. Why does LC3 have obvious distribution in zona pellucida, and whether the specificity of antibody is poor? What form and distribution does the small green spot represent for LC3?

6. Fig6. The staining seems to be nonspecific, the increase of Nrf2 is not obvious, and the change of gene expression is not completely consistent with that described in the result. It is suggested to carry out WB experiment to verify the change of Nrf2 expression.

7. There was no error bar in qPCR control group, and the statistical difference was not indicated. Generally, it is more meaningful when gene expression changes more than twice in qPCR detection.

8. Please discuss the regulation of pluripotent genes in early embryos and their relationship with cell proliferation.

9. Early embryonic development requires a certain degree of autophagy, which is conducive to the elimination of maternal factors and the reduction of ROS. Please discuss it in combination with the results of this study.

10. Line286-292. Please check the content and logic, including the relationship between autophagy, p62 and Nrf2. According to the results of this study, autophagy decreased rather than increased after WDL treatment. In most autophagy processes, the abnormal increase of autophagy will be accompanied by the decrease of substrate p62.

Minor points
1、Line11 Please check the description “anti-free radical scavenging”

2、Line162-163 Please check the PCR condition of “37℃ 30s”; should be 18s rRNA

3、Line147 Please indicate the specific time of “overnight”

4、Line169 The significance of difference (P < 0.05) was not defined

5、Line 180 remove “%”

---

## Round 0.2 · Minor Revisions

Dear Authors,
In reviewing the information provided by the reviewers, I am requesting revisions before the article can be considered further.

Reviewer 1 ·

Basic reporting

The article must be written in English and must use clear, unambiguous, technically correct text. The article must conform to professional standards of courtesy and expression.

Experimental design

The submission should clearly define the research question, which must be relevant and meaningful. The knowledge gap being investigated should be identified, and statements should be made as to how the study contributes to filling that gap.

Validity of the findings

The conclusions should be appropriately stated, should be connected to the original question investigated, and should be limited to those supported by the results (the concentration of WDL).

Additional comments

Introduction : should have the information of WDL concentrations that studied in cell or embryo
Line 32 &254 : in vivo, for this word is no need to italic letter
Line 64-65 : should change to Alpha-Ketoglutarate
Line 153 : move for 1 h to the end of sentence (please check other sentence too)
Line 174 : should change to Ten embryos
Line 208 : organic cation transporter 4 or octamer binding transcription factor 4
Line 334 : In conclusion should have the information of WDL concentration
For figure legends : in figure 1 : very confuse between superscripts A, B, C and a,b,c in day 6 and day 7 in figure 1B and figure 1A 1B and 1C. For the suggestion, should change the superscript symbols.

---

## Round 0.3 · Minor Revisions

Dear Authors,

Thank you very much for resubmitting the modified version of the manuscript. However, there are at least a few minor things to correct:

Line 41: the systematic name for wedelolactone is wrong: 7-methoxy-5,11,12-trihydroxybenzene does not describe any possible chemical entity (since benzene only has 6 carbon atoms and cannot, therefore, have substituents n carbons 11 or 12. The correct IUPAC name for this molecule is, if one is allowed to trust Wikipedia, 1,8,9-Trihydroxy-3-methoxy-6H-[1]benzofuro[3,2-c][1]benzopyran-6-one

The protocol for the parthenogenetic activation lacks referencing. Even if it is a new protocol, some information regarding the rationale behind each step (e.g. referencing previous protocols used for inspiration) should have been provided.

In line 101, the Richani et al.2021 reference is out-of-place: that reference is a review on oocyte metabolism, not a paper that describes the separation of oocytes from cumulus cells, or other methodological details."

Please make necessary amendments accordingly.

---

## Round 0.4 · Minor Revisions

Thank you very much for resubmitting the modified version of the manuscript. However, the Section Editor has commented:

"The authors have not yet added the requested references to the experimental method for parthenogenetic activation"

Please add the reference for the parthenogenetic activation method.

Thank you

---

## Round 0.5 · accepted · Accept

The comment has been addressed and I suggest the acceptance of this manuscript.